# Adsorption of Heavy Metal Ions Copper, Cadmium and Nickel by *Microcystis aeruginosa*

**DOI:** 10.3390/ijerph192113867

**Published:** 2022-10-25

**Authors:** Guoming Zeng, Yu He, Dong Liang, Fei Wang, Yang Luo, Haodong Yang, Quanfeng Wang, Jiale Wang, Pei Gao, Xin Wen, Chunyi Yu, Da Sun

**Affiliations:** 1Chongqing Engineering Laboratory of Nano/Micro Biological Medicine Detection Technology, School of Architecture and Engineering, Chongqing University of Science and Technology, Chongqing 401331, China; 2Intelligent Building Technology Application Service Center, Chongqing City Vocational College, Chongqing 402160, China; 3Department of Construction Management and Real Estate, Chongqing Jianzhu College, Chongqing 400072, China; 4Institute of Life Sciences & Biomedical Collaborative Innovation Center of Zhejiang Province, National & Local Joint Engineering Research Center for Ecological Treatment Technology of Urban Water Pollution, Wenzhou University, Wenzhou 325000, China

**Keywords:** *Microcystis aeruginosa*, heavy metal, removal rate, structural analysis

## Abstract

To investigate the treatment effect of algae biosorbent on heavy metal wastewater, in this paper, the adsorption effect of *M. aeruginosa* powder on heavy metal ions copper, cadmium and nickel was investigated using the uniform experimental method, Fourier transform infrared spectroscopy (FTIR), scanning electron microscopy (SEM) and TG-DSC comprehensive thermal analysis. The experimental results showed that the initial concentration of copper ion solution was 25 mg/L, the temperature was 30 °C, the pH value was 8 and the adsorption time was 5 h, which was the best condition for the removal of copper ions by algae powder adsorption, and the removal rate was 83.24%. The initial concentration of cadmium ion solution was 5 mg/L, the temperature was 35 °C, the pH value was 8 and the adsorption time was 4 h, which was the best condition for the adsorption of cadmium ion by algae powder, and the removal rate was 92.00%. The initial nickel ion solution concentration of 15 mg/L, temperature of 35 °C, pH value of 7 and adsorption time of 1 h were the best conditions for the adsorption of nickel ions by algae powder, and the removal rate was 88.67%. The spatial structure of algae powder changed obviously before and after adsorbing heavy metals. The functional groups such as amino and phosphate groups on the cell wall of *M. aeruginosa* enhanced the adsorption effect of heavy metal ions copper, cadmium and nickel. Additionally, *M. aeruginosa* adsorption of heavy metal ions copper, cadmium, nickel is an exothermic process. The above experiments show that *M. aeruginosa* can be used as a biological adsorbent to remove heavy metals, which lays a theoretical foundation for the subsequent treatment of heavy metal pollution by algae.

## 1. Introduction

With the rapid development of industry and agriculture, heavy metal wastewater is discharged into natural water bodies in large quantities, which poses a great threat to the ecological environment. Because heavy metals cannot be degraded naturally, once they enter the food chain, they will have a significant impact on human life and health [1]. Excessive copper intake in the human body can cause poisoning: hepatolenticular degeneration and cholestasis in children are all symptoms of copper poisoning. After cadmium is absorbed by the human body, cadmium protein is formed in the body, which is easily stored in the kidney and liver. Cadmium poisoning can easily damage the normal function of kidney and liver organs, hinder the metabolism of human bones, and cause a series of orthopedic diseases. Nickel can easily cause allergic dermatitis after enrichment in the human body, and nickel has carcinogenic effects [2,3]. Therefore, the effective removal of heavy metal ions in wastewater is an urgent environmental problem to be solved. Conventional governance methods are mainly divided into chemical methods, physical methods and biological methods. The chemical precipitation method, ion exchange method, activated carbon adsorption method and other physical and chemical methods have their own advantages, but the investment is large and can easily produce secondary pollution problems; the biological method uses biological materials as adsorbents to treat heavy metal-contaminated water bodies, which is not only green and environmentally friendly, but also economical and economical. It is in line with the current environmental protection concept advocated by the country and has a good development prospect. It is also a hot topic in current research [4,5,6]. Materials that can be used as biosorbents mainly include bacteria, fungi, yeast, microalgae, etc. [7,8]. Algae has a good adsorption effect on heavy metals in water due to its porous structure, large surface area, wide distribution, short growth cycle, safety and environmental protection [9,10,11].

In recent years, Chen et al. [12] studied the adsorption effect of immobilized *M. aeruginosa* on Pb^2+^, Cd^2+^ and Hg^2+^, and found that *M. aeruginosa* had high affinity for the adsorption of Pb^2+^, Cd^2+^ and Hg^2+^. When the biosorption reached a steady state, the removal efficiency of Cd^2+^ and Hg^2+^ reached 90%, and the removal efficiency of Pb^2+^ reached 80%. Tao et al. [13] studied the factors affecting the adsorption of Cd^2+^ and Pb^2+^ by suspended *M. aeruginosa*. It was found that the removal rate of Pb^2+^ was 90–100% and Cd^2+^ was 79.5–100% after 3 h and 6 h of culture at pH = 7, temperature 25 °C and light conditions. Zeng et al. [14] found that the concentration of Cd^2+^ in algal cells increased with the increase in exposure time, while the concentration of Zn^2+^ in cells always reached a steady state by studying the bioaccumulation characteristics of Cd^2+^ and Zn^2+^ in *M. aeruginosa*. After long-term exposure to different concentrations of free Cd^2+^ or Zn^2+^, the change in Cd^2+^ was greater than that of Zn^2+^. Deng et al. [15] studied the physiological response of *M. aeruginosa* to Zn^2+^ and Cd^2+^ and their ability to accumulate ions and found that low concentrations (less than 0.1 mg/L) of Zn^2+^ and Cd^2+^ had little effect on the growth and physiological processes of algae, while higher concentrations (more than 0.1 mg/L) significantly inhibited the cell division of *M. aeruginosa* (from 12.6% to 70.0%) and photosynthetic performance (from 7.1% to 53.1%). The accumulation of Zn^2+^ or Cd^2+^ by *M. aeruginosa* increased exponentially with the initial concentration of metal ions. Ni et al. [16] studied the absorption of cadmium by *M. aeruginosa* in the presence of dissolved organic matter (DOM) from different sources. It was found that the molecular weight of DOM in sediments and soil in Meiliang Bay of Taihu Lake was lower than that of *M. aeruginosa*. When exposed to Cd^2+^, the EEM fluorescence intensity of the three DOMs decreased significantly, and the intracellular Cd^2+^ content increased with the increase in Cd^2+^ concentration. Compared with the control group, the addition of DOM greatly enhanced the absorption of Cd^2+^ by *M. aeruginosa*. Rzymski et al. [17] studied the bioaccumulation levels of Cd^2+^ and Pb^2+^ in *M. aeruginosa* and found that under the stress of higher concentrations of 20 mg/L Cd and 10–20 mg/L Pb, *M. aeruginosa* showed higher bioaccumulation of Cd and Pb, which was proportional to the initial metal concentration. 

In this paper, *M. aeruginosa* was used to remove heavy metal ions. The optimum conditions for the treatment of three heavy metal ions by *M. aeruginosa* were obtained by uniform experimental design. Fourier transform infrared spectroscopy (FTIR), scanning electron microscopy (SEM) and thermogravimetric analysis-differential scanning calorimetry (TG-DSC) were used to analyze the changes in intracellular functional groups, surface morphology and thermal energy of algal cells before and after the adsorption of heavy metal ions copper, cadmium and nickel by *M. aeruginosa*. The results of this study can provide theoretical parameters for the study of heavy metal adsorption by algae.

## 2. Materials and Methods

### 2.1. Algal Species 

*M. aeruginosa* was purchased from Algae Preservation Center, Institute of Aquatic Biology, Chinese Academy of Sciences. The culture medium was provided by the Algae Preservation Center, Institute of Aquatic Biology, Chinese Academy of Sciences.

Culture conditions: temperature (25 ± 1) °C, light intensity 2000lx~2500lx, light to dark ratio of 12 h:12 h, static culture, shaking regularly 3 times a day, so that the same culture medium algae cell growth tends to be at the same stage and microscopic examination of cells after normal experiments.

### 2.2. Preparation of Algal Powder

With a 100 mL beaker to take 50 mL algae (in the logarithmic phase of growth), suction filtration, algae attached to the filter membrane, and then placed in a drying oven, and finally with a blade from the algae scraped off the filter membrane, weighed 5 g.

### 2.3. Adsorption Experiment

Anhydrous copper sulfate, nickel chloride hexahydrate and cadmium nitrate were dissolved in distilled water to obtain a metal ion solution, and then the algae powder was added to a conical flask containing a certain concentration of heavy metal solution, fully mixed and oscillated. After adjusting the pH value, it was placed in a constant temperature incubator to set the adsorption time and temperature. Through the uniform design software (DPS), the pH value (4,5,6,7,8), temperature (15 °C, 20 °C, 25 °C, 30 °C, 35 °C), metal concentration (5 mg/L, 10 mg/L, 15 mg/L, 20 mg/L, 25 mg/L), reaction time (1 h, 2 h, 3 h, 4 h, 5 h) were uniformly designed to obtain 12 test schemes, as shown in Table 1. The removal experiments of cadmium, nickel and copper were carried out according to the conditions in Table 1.

### 2.4. Removal Rate

The mixed liquid at the end of the adsorption experiment was filtered to obtain the concentration of the supernatant after adsorption. Then, 10 mL was taken in a centrifuge tube and detected by atomic absorption spectrometer. The removal rate of heavy metals was calculated by Equation (1).
(1)$$\mathrm{Removal}\;\mathrm{rate}=\frac{C-C_{0}}{C_{0}}\times 100\%$$
where *C*_0_ represents the concentration of heavy metals before reaction (mg/L); *C* represents the concentration of heavy metals after reaction (mg/L) mixed.

### 2.5. Analysis Method

Infrared spectrum analysis: KBr pellet method, instrument model: Avatar-360 infrared spectrometer (USA). The spectral width is 4000–400 cm^−1^. Scanning electron microscopy analysis: SEM acceleration voltage is 10 kV, magnification is 1000 times. The test sample was determined by a thermogravimetric/differential thermal analyzer. The model was STA409, the sample weight was about 10 mg, Al_2_O_3_ was used to calibrate the temperature, N_2_ protection, and the test temperature range was from room temperature to 1000 °C. The BET method was used to determine the specific surface area of *M. aeruginosa* powder, and the BJH model was used to calculate the pore diameter and pore volume.

### 2.6. Statistical Analysis Method

The experimental data were analyzed and processed by the Origin 8.0 software processing system and Windows Excel, Word (2003, 2010 edition) office software.

## 3. Results

### 3.1. Optimization Analysis of Uniform Experiment Process

According to the uniform design experimental scheme, the experiments of *M. aeruginosa* adsorbing copper ions, cadmium ions and nickel ions were carried out respectively. The experimental results are shown in Table 2.

The BET method showed that the specific surface area of *M. aeruginosa* powder was 6.59 m^2^/g, the BJH method showed that the pore volume of *M. aeruginosa* powder was 0.023 cm^3^/g, and the average pore size was 3.145 nm. The *M. aeruginosa* cell wall has a large number of functional groups such as hydroxyl, carboxyl, amino and amide groups [18], and its particle size is small and has micropores, which has a good adsorption capacity for heavy metals. It can be used as a model algae for the removal of heavy metals. The following conclusions can be drawn from Table 1 and Table 2: the initial copper ion solution was 25 mg/L, the temperature was 30 °C, the pH value was 8 and the adsorption time was 5 h. Under this condition, the removal rate of copper ions in water by *M. aeruginosa* reached 83.24%. When the cadmium ion solution was 5 mg/L, the temperature was 35 °C, the pH value was 8 and the adsorption time was 4 h and the removal rate of cadmium ion in water by *M. aeruginosa* reached 92.00%. When the nickel ion solution was 15 mg/L, the temperature was 35 °C, the pH value was 7 and the adsorption time was 1 h; the removal rate of nickel ions in water by *M. aeruginosa* reached 88.67%. According to related studies, algae absorption of heavy metals was found to occur in two steps: the first step is the passive adsorption process (physical adsorption or ion exchange). This adsorption process is rapid: heavy metals are simply bound to the surface of algae cells. The second step is the active absorption process, which is the main way for algae cells to absorb heavy metal ions. The absorption rate of heavy metals by algae cells is related to the concentration of heavy metal ions, temperature, pH and adsorption time. Therefore, the adsorption process of heavy metal ions copper, cadmium and nickel by *M. aeruginosa* is a complex physical and chemical process affected by multiple factors. It can effectively adsorb copper ions, cadmium ions and nickel ions in water under appropriate conditions.

### 3.2. FTIR Analysis

As shown in Figure 1, *M. aeruginosa* powder showed characteristic absorption bands in the following spectral regions: (1) 3500~3400 cm^−1^: the maximum absorption peak is near 3435.06 cm^−1^, strong and wide, which is the stretching vibration peak of O-H and N-H bonds in cellular polysaccharides and proteins; (2) the maximum absorption peak at 3000–2900 cm^−1^ was near 2925.28 cm^−1^, and the peak intensity was moderate, which was caused by the asymmetric stretching vibration of C-H in the alkyl groups of proteins and carbohydrates on the cell wall [19,20]; (3) the stretching peak at 1600~1500 cm^−1^: 1642.10 cm^−1^ is caused by the C = O vibration in the amide I (CO-NH) of the alkyl group in the protein and sugar on the cell wall; (4) vibration caused by-COOH at 1400~1300 cm^−1^: 1381.12 cm^−1^; (5) 1100~1000 cm^−1^: C-O and P-O (P = O) on peptidoglycan, polysaccharide and phospholipid in cells caused a stretching vibration at 1054.23 cm^−1^ [21,22]. The infrared spectrum showed that although the peak shape of the four heavy metal ions copper, cadmium and nickel was basically unchanged, the peak of the algae powder changed compared with the original sample. After the algae powder absorbed copper, cadmium and nickel ions, the peaks moved from 3435.06 cm^−1^ to 3425.94 cm^−1^, 3407.43 cm^−1^, 3420.07 cm^−1^, 2925.28 cm^−1^ to 2927.09 cm^−1^, 2927.31 cm^−1^, 2926.12 cm^−1^, respectively, indicating that the O-H, N-H and C-H groups contained in proteins and sugars in cells were involved in the adsorption process. The shift of 1642.10 cm^−1^ to 1650.97 cm^−1^, 1655.57 cm^−1^, 1657.81 cm^−1^ indicated that the amide in the cell also participated in the reaction. The shift of 1381.12 cm^−1^ to 1386.22 cm^−1^, 1398.96 cm^−1^, 1391.51 cm^−1^ indicated that the-COOH group in the cell played an important role in the adsorption of nickel ions. The shift of 1054.23 cm^−1^ to 1060.71 cm^−1^ and 1048.83 cm^−1^, respectively, 1075.76, was caused by C-O and P-O (P = O) stretching vibration, indicating that peptidoglycan, polysaccharides and phospholipids in cells were involved in the reaction process of nickel ions [23,24,25]. In summary, *M. aeruginosa* adsorbs heavy metals.

### 3.3. SEM Analysis

The SEM images of *M. aeruginosa* before and after adsorption are shown in Figure 2. The surface structure of *M. aeruginosa* is uneven, the structure is porous, and the surface area is large, which is very conducive to its adsorption of heavy metals, because this will expose many functional groups to provide sufficient adsorption space [26,27]. These functional groups can be reasonably arranged on the algal cell wall with a large surface area to fully contact with metal ions. Some of them can lose protons and have negative charges and adsorb metal ions by electrostatic attraction; some with lone pair electrons can form coordination bonds with metal ions and complex adsorption of metal ions. At the same time, the cell wall also has a certain charge and viscosity, which increases its adsorption capacity for metal ions. Among all functional groups, the carboxyl group provided by polysaccharides is the most important [28,29,30]. The morphological structure of *M. aeruginosa* changed obviously before and after treatment. Before treatment, the surface of the original structure has less attachment and larger pore size. After treatment, it can be clearly observed that many particulate matter attached to the surface of *M. aeruginosa*, pore size becomes smaller and the structure becomes disorderly. This is consistent with the results of the previous infrared analysis. The results show that the algae has a certain effect on the adsorption of heavy metal ions [31,32].

### 3.4. TG-DSC Analysis

As an important means of characterization, thermal analysis is a method to analyze the composition and thermal stability of the tested materials by temperature programming, causing changes in their physical and chemical properties to obtain a series of data. Differential scanning calorimetry (DSC) and thermogravimetry (TG) are two commonly used thermal analysis methods. TG is a method for analyzing the relationship between the mass and temperature of the sample to be tested, while DSC is a method for analyzing the relationship between the endothermic or exothermic rate of the sample to be tested and the temperature [33,34]. Figure 3 is the TG-DSC curve of the absorption of copper, cadmium and nickel ions by *Microcystis aeruginosa*. The TG changes in Figure 3a–c show that the absorption of heavy metal ions by *Microcystis aeruginosa* has three obvious weight loss processes: weight loss at about 100 °C, mainly due to the volatilization of residual water in algae powder; weight loss at 200~400 °C is mainly caused by the combustion of easily degradable substances and semi-volatile components (such as carbohydrates, aliphatic structures, carboxyl groups, etc.) and hemicellulose, cellulose, microbial cell wall and other substances. A total of 400~600 °C weight loss, mainly due to the high molecular weight of aromatic and polycyclic structures, such as lignin, humic acid substances caused by the combustion. As shown in Figure 3a–c, the DSC curve of *M. aeruginosa* absorbing copper, cadmium and nickel ions has an obvious endothermic and exothermic peak at 500~900 °C, and with the increase in heating rate, the peak moves to the right and moves in the high temperature direction. This indicates that the adsorption of copper, cadmium and nickel ions by *M. aeruginosa* powder leads to the formation of an exothermic peak on the DSC curve. Adsorption of heavy metal ions copper, cadmium and nickel after the energy is greater than before adsorption; the algae cell adsorption process is an endothermic reaction [35,36].

## 4. Conclusions

(1) Through the uniform test, it can be seen that the removal rate of copper ions in water by *M. aeruginosa* reached the highest 83.24% under the conditions of copper ion solution concentration of 25 mg/L, temperature of 30 °C, pH value of 8 and adsorption time of 5 h. Under the conditions of cadmium ion solution 5 mg/L, temperature 35 °C, pH value 8 and adsorption time 4 h, the removal rate of cadmium ion in water by *M. aeruginosa* reached the highest, which was 92.00%. Under the conditions of nickel ion solution of 15 mg/L, a temperature of 35 °C, pH value of 7 and adsorption time of 1 h, the removal rate of nickel ion in water by *M. aeruginosa* reached the highest, which was 88.67%.

(2) The results of FIIR and SEM showed that the functional groups such as amino, phosphate, hydroxyl, carboxyl and carbonyl on the cell wall of *M. aeruginosa* enhanced the treatment effect of heavy metal ions copper, cadmium and nickel. The spatial structure and morphology of algae cells changed significantly before and after treatment, indicating that *M. aeruginosa* had a certain adsorption effect on heavy metal ions copper, cadmium and nickel.

(3) TG-DSC analysis showed that the process of *M. aeruginosa* adsorbing heavy metal ions copper, cadmium and nickel was an exothermic process.

(4) *M. aeruginosa*, as a relatively large algae in water bloom, has an extremely low cost and certain adsorption effects on copper ion, cadmium ion and nickel ion. It is expected to become a biological adsorbent for treating wastewater containing heavy metal ions copper, cadmium and nickel. Therefore, its successful application in the removal of heavy metal ions in wastewater has broad development prospects.

## Figures and Tables

**Figure 1 ijerph-19-13867-f001:**
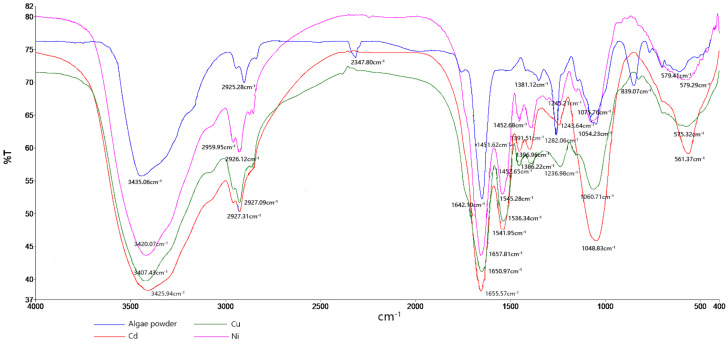
Infrared spectrum of *Microcystis aeruginosa* before and after adsorption of heavy metal ions Cu^2+^, Cd^2+^, Ni^2+^.

**Figure 2 ijerph-19-13867-f002:**
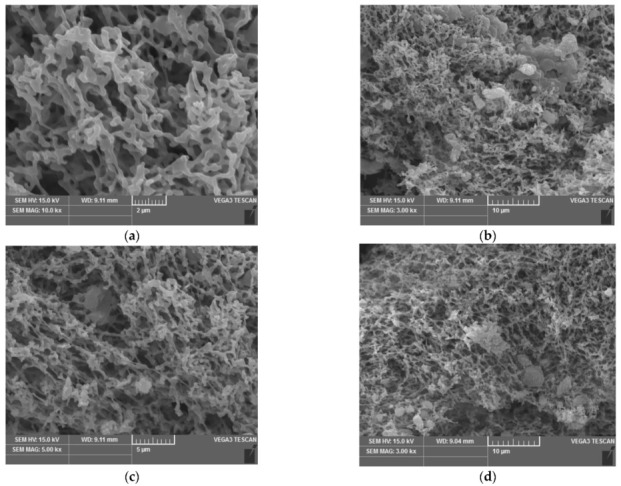
The SEM images of *Microcystis aeruginosa* before and after adsorbing copper, cadmium and nickel ions. (**a**) Algae powder; (**b**) Adsorption of Cu^2+^ by algae powder; (**c**) Adsorption of Cd^2+^ by algae powder; (**d**) Adsorption of Ni^2+^ by algae powder.

**Figure 3 ijerph-19-13867-f003:**
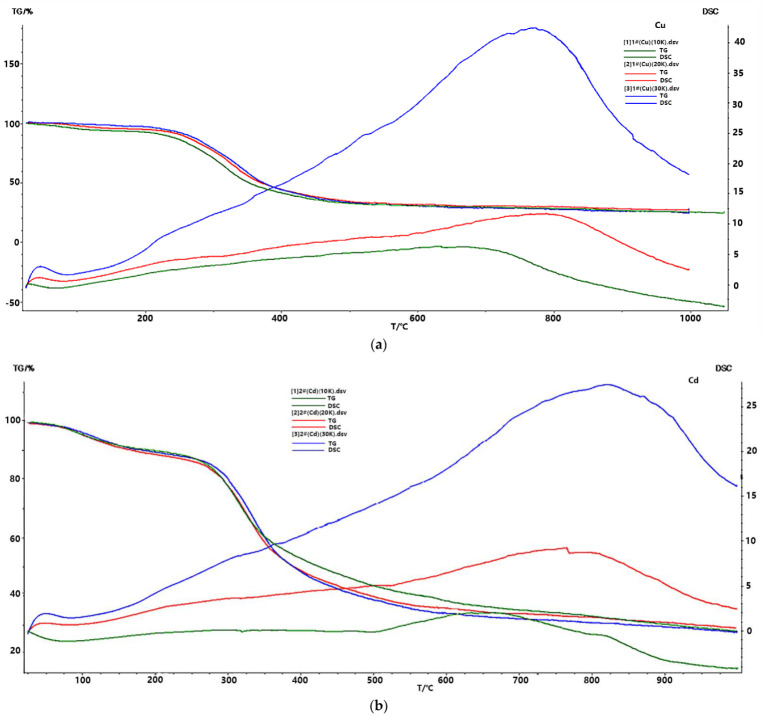
TG-DSC diagram of *Microcystis aeruginosa* before and after adsorption of copper, cadmium and nickel ions. (**a**) TG-DSC diagram of *Microcystis aeruginosa* before and after adsorption of copper ions; (**b**) TG-DSC diagram of *Microcystis aeruginosa* before and after adsorption of cadmium ions; (**c**) TG-DSC diagram of *Microcystis aeruginosa* before and after adsorption of nickel ions, The green, red and blue lines represent the heating rates of 10 K/min, 20 K/min and 30 K/min, respectively.

**Table 1 ijerph-19-13867-t001:** Different algae powder adsorption experimental conditions (pH, temperature, adsorption time, initial heavy metal concentration).

Variable Sequence	Mass Concentration of Heavy Metals (mg/L)			Temperature (°C)	pH	Adsorption Time (h)
	Cu^2+^	Cd^2+^	Ni^2+^			
1	20	20	20	15	6	1
2	15	15	15	35	7	1
3	5	5	5	30	4	2
4	25	25	25	20	4	4
5	10	10	10	25	5	5
6	10	10	10	20	8	3
7	15	15	15	15	7	5
8	25	25	25	30	8	5
9	25	25	25	35	5	3
10	5	5	5	35	8	4
11	20	20	20	35	6	5
12	25	25	25	25	8	2

**Table 2 ijerph-19-13867-t002:** Removal Rate of Copper, Cadmium and Nickel by Algae Powder.

Variable Sequence	Cu^2+^			Cd^2+^		Ni^2+^	
	Initial Metal Concentration	Final Metal Concentration	Removal Rate	Final Metal Concentration	Removal Rate	Final Metal Concentration	Removal Rate
1	20	17.01	14.95%	15.60	22.00%	19.40	3.00%
2	15	13.10	12.67%	3.67	75.53%	1.70	88.67%
3	5	4.94	1.20%	2.10	58.00%	4.90	2.00%
4	25	19.69	21.24%	19.00	24.00%	20.80	16.80%
5	10	8.98	10.20%	7.60	24.00%	8.60	14.00%
6	10	9.49	5.10%	2.50	75.00%	8.50	15.00%
7	15	13.56	9.60%	7.70	48.66%	12.50	16.67%
8	25	4.19	83.24%	5.06	79.76%	18.40	26.40%
9	25	22.93	8.28%	16.60	33.60%	24.10	3.60%
10	5	3.62	27.60%	0.40	92.00%	4.80	4.00%
11	20	4.92	75.40%	9.10	54.50%	15.90	20.50%
12	25	22.47	10.12%	3.60	85.60%	22.10	11.60%

## Data Availability

Not applicable.

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
