# Peer review of "Adsorption of Heavy Metal Ions Copper, Cadmium and Nickel by Microcystis aeruginosa"

_ijerph, 2022, doi:10.3390/ijerph192113867_

Round 1

Reviewer 1 Report

The article is scientifically sounding however there is still a room for improvement.

Title
The title is not accurately reflect the study. It should accurately describe
the study. for instead it must include the method used in the study. This title sound as it if for a review paper not for an article.

Abstract

The first statement is incomplete. The abstract does not follow the required format. The abstract does not contain information absent in the main text?
The author must visit the language editor so that the results in the abstract can be presented properly.
Introduction
The are no adequate background information from literature describe the why Microcystis aeruginosa as an adsorber. No information whether others  have carried out similar studies. language grammar must be checked
Methods/Experimental
All the methods  employed in this study are not adequately described. the author have to describe adequately all the reagent  used in the preparation of the metal  ion solutions  (which salts were used to prepare the assays).. Were proper validation experiments performed?
where applicable, how was the samples collected, processed, and stored?
language must be checked.

Results

The results presented are unbalanced and biased and authors do not provide adequate data to support the results. SEM, TG-DSC and IR are insufficient to arrived to the given conclusion. For instead, the anions such as Cl-, SO42- can affect the spectra of both TG-DSC and IR and couse the chages in the orientation of the spectra. BET, Iodine number and methylene number can be one of the methods that can be used to confirm the porous and adsorption of metal ions on the algae powder.  The figures and tables are not well labelled and are not clear

References

The authors cite unpublished materials. example 4, 13 and 19

Reviewer 2 Report

Well done for your great work. Kindly perform the improvement as follows before potential acceptance:

1.       The abstract seems abit too long, please remove the less important messages.

2.       Please correct the in-text citation format in the whole mansucript. The author should check the guide for author properly.

3.       Please highlight the novelty in the introduction.

4.       Please redraw the Figure 1, do not use raw graphs.

5.       Poor resolution quality for Figure 2(b), please revise.

6.       Please redraw the Figure 3, do not use raw graphs. 

Reviewer 3 Report

The subject investigated in this manuscript is interesting and the experiments seem adequate to achieve the objectives. The methodology is not complete and results should be presented more clearly. The conclusions should be revised.

General comments:

Microcystis aeruginosa is the scientific name. In cursive!! revise the scientific names through the manuscript

Always include initial concentration when inform about removal percentage. Otherwise, the information is not complete

Methodology: what is the volume of metal solution used in the experiments? And algae mass?.

What method and instrument were used to measure metal concentration? Were the samples acidified before analysis?

Table 1. Algae powder protocol. This table show experimental conditions. Change the caption.

What was the software to plan the experimental design? Justify the range of experimental conditions selection (temperature, pH time, concentration,…)

Table 2 Calculation table of removal rate of copper, cadmium and nickel treated with algae liquid . Remove calculation table. What is the meaning of algae liquid?

Heavy metal Constituencies? it is the number of experiment related to table 1!!!

Before adsorption and After adsorption column?? Initial and final metal concentration are more appropriated. Include units ( mg/L).

Check number of significant figures on removal percentage (22.000%???).

The discussion of table 1 and 2 is based on the experimental conditions of higher removal percentage. Anything else? too poor!!

 Figure 2. (a) Before algae powder treatment. What treatment?  

(b) Adsorption of Cu2+ by algae powder;(c) Adsorption of Cd2 +by algae powder; (d) Adsorption of Ni2+ by algae powder. Adsorption can not be seen in a picture!! Change the figure legend to inform about what you present.

Include information about sorption conditions of SEM samples

Include references to support the discussion from line 203 to line 207.

 What are the observed particles attached to the surface of Microcystis aeruginosa? A chemical analysis of these particles ( SEM-EDX?) sould be done to know the composition. It seem that metals precipitation could play an important role in this metal removal ( Cu2+ at 5 mg/L, and higher concentrations, is present as CuO in aqueous solutions at pH> 5.8. What is the final pH of solutions? ( after sorpton process). Please, revise the discussion. 

Figure 3. (a) Adsorption of Cu2+ by Microcystis aeruginosa; (b) Adsorption of Cd2 by Microcystis aeruginosa;(c) Adsorption of Ni2 +by Microcystis aeruginosa. Adsorption can not be seen in a diagram!! Change the figure legend to inform about what you present.

 What is the meaning of “had good adsorption effects” in conclusion 2 and 4?

Round 2

Reviewer 1 Report

The results presented are unbalanced and biased and authors do not provide adequate data to support the results.  BET or Iodine number or/and methylene number must be included to confirm the porous and adsorption of metal ions on the algae powder.  

Author Response

Dear editor,

Thank you for your good comments and suggestions. The BET test and BJH test were conducted. BET method was used to determine the specific surface area of M. aeruginosa powder, and BJH model was used to calculate the pore diameter and pore volume, please check it.

Warm regards

Guoming Zeng

Reviewer 3 Report

The most important points have been corrected accordingly to the recommendations. Some other aspects has not been clarified (authors said “due to the impact of the epidèmic”) , even though they would help to validate the hypothesis and improve the scientific quality of this manuscript.   Nevertheless, after the modifications done, I consider this manuscript can be published in a scientific journal

Author Response

Dear editor ,

Thank you for your understanding and support. In order to make the article more complete and read by the researcher smoothly, we added BET and BJH test data. BET method was used to determine the specific surface area of M. aeruginosa powder, and BJH model was used to calculate the pore diameter and pore volume, please check it.

Warm regards

Guoming Zeng
